# Vocal Call Locator Benchmark (VCL) for localizing rodent vocalizations from multi-channel audio

**Ralph E Peterson**[1,2,*,†], **Aramis Tanelus**[2,*], **Christopher Ick**[3], **Bartul Mimica**[4], **Niegil Francis**[1,5],
**Violet J Ivan**[1], **Aman Choudhri**[6], **Annegret L Falkner**[4], **Mala Murthy**[4],
**David M Schneider**[1], **Dan H Sanes**[1], **Alex H Williams**[1,2,†]

[1]NYU, Center for Neural Science
[2]Flatiron Institute, Center for Computational Neuroscience
[3]NYU, Center for Data Science
[4]Princeton Neuroscience Institute
[5]NYU, Tandon School of Engineering
[6]Columbia Univsersity

[*]Equal contribution
[†]Correspondence to `rep359@nyu.edu` and `alex.h.williams@nyu.edu`

## Abstract

Understanding the behavioral and neural dynamics of social interactions is a goal of contemporary neuroscience. Many machine learning methods have emerged in recent years to make sense of complex video and neurophysiological data that result from these experiments. Less focus has been placed on understanding how animals process acoustic information, including social vocalizations. A critical step to bridge this gap is determining the senders and receivers of acoustic information in social interactions. While sound source localization (SSL) is a classic problem in signal processing, existing approaches are limited in their ability to localize animal-generated sounds in standard laboratory environments. Advances in deep learning methods for SSL are likely to help address these limitations, however there are currently no publicly available models, datasets, or benchmarks to systematically evaluate SSL algorithms in the domain of bioacoustics. Here, we present the VCL Benchmark: the first large-scale dataset for benchmarking SSL algorithms in rodents. We acquired synchronized video and multi-channel audio recordings of 767,295 sounds with annotated ground truth sources across 9 conditions. The dataset provides benchmarks which evaluate SSL performance on real data, simulated acoustic data, and a mixture of real and simulated data. We intend for this benchmark to facilitate knowledge transfer between the neuroscience and acoustic machine learning communities, which have had limited overlap.

**Data is available at:** `vclbenchmark.flatironinstitute.org`

## 1  Introduction

An ongoing renaissance of ethology in the field of neuroscience has shown the importance of conducting experiments in naturalistic contexts, particularly social interactions [40, 1]. Most experiments in social neuroscience have focused on relatively constrained contexts over short timescales, however an emerging paradigm shift is leading laboratories to adopt longitudinal experiments in semi-natural or natural environments [52]. With this shift comes significant data analytic challenges—such as how to track individuals in groups of socially interacting animals—necessitating collaboration between the fields of machine learning and neuroscience [10, 46].

38th Conference on Neural Information Processing Systems (NeurIPS 2024) Track on Datasets and Benchmarks.

Substantial progress has been made in applying machine vision to multi-animal pose tracking and action recognition [47, 33, 38, 58], however applications of machine audio for acoustic analysis of animal generated social sounds (e.g. vocalizations or footstep sounds) have only recently begun [51, 20]. To study the dynamics of vocal communication and their neural basis, ethologists and neuroscientists have developed a multitude of approaches to attribute vocal calls to individual animals within an interacting social group, however many existing approaches for vocalization attribution necessitate specialized experimental apparatuses and paradigms that hinder the expression of natural social behaviors. For example, invasive surgical procedures, such as affixing custom-built miniature sensors to each animal [17, 50, 64], are often needed to obtain precise measurements of which individual is vocalizing. In addition to being labor intensive and species specific, these surgeries are often not tractable in very small or young animals, may alter an animal's natural behavioral repertoire, and are not scalable to large social groups. Thus, there is considerable interest in developing non-invasive sound vocal call attribution methods that work off-the-shelf in laboratory settings.

Sound source localization (SSL) is a decades old problem in acoustical signal processing, and several neuroscience groups have adapted classical algorithms from this literature to localize animal sounds [43, 56, 66]. These approaches can work reasonably well in specialized acoustically transparent environments, however they tend to fail in reverberant environments (see Supplement) that are required for next-generation naturalistic experiments.

Data-driven modeling approaches with fewer idealized assumptions may be expected to achieve greater performance [69]. Indeed, in the broader audio machine learning community, deep networks are commonly used to localize sounds [22]. Typically, these approaches have been targeted at human-scale acoustic environments—e.g. localizing sounds within rooms of a home [54]. To our knowledge, no benchmark datasets or deep network models have been developed for localizing sounds emitted by small animals (e.g. rodents) interacting in common laboratory environments (e.g. a spatial footprint less than one square meter). To address this, we present benchmark datasets for training and evaluating SSL techniques in reverberant conditions.

## 2 Background and Related Work

### 2.1 Existing Benchmarks

Acoustic engineers are interested in SSL algorithms for a variety of downstream applications. For example, localization can enable audio source separation [36] by disentangling simultaneous sounds emanating from different locations. Other applications include the development of smart home and assisted living technologies [19], teleconferencing [65], and human-robot interactions [34]. To facilitate these aims, several benchmark datasets have been developed in recent years including the L3DAS challenges [23, 24, 21], LOCATA challenge [15], and STARSS23 [54].

Notably, all of these applications and associated benchmarks are (a) focused on a range of sound frequencies that are human audible, and (b) focused on large environments such as offices and household rooms with relatively low reverberation. Our benchmark differs along both of these dimensions, which are important for neuroscience and ethology applications.

Many rodents vocalize and detect sounds in both sonic and ultrasonic ranges. For example, mice, rats, and gerbils collectively have hearing sensitivity that spans ~50-100,000 Hz with vocalizations spanning ~100-100,000 Hz [45]. Localizing sounds across a broad spectrum of frequencies introduces interesting complications to the SSL problem. Phase differences across microphones carry less reliable information for higher frequency sounds (see e.g. [29]). Moreover, a microphone's spatial sensitivity profile will generally be frequency dependent (see microphone specifications for ultrasonic condenser microphone CM16-CMPA from Avisoft Bioacoustics). Therefore, sounds emanating from the same location with the same source volume but distinct frequencies can register with unique level difference profiles across microphones. Thus, different acoustical computations are required to perform SSL for high and low frequency sounds. Indeed, we find that deep networks trained on low frequency sounds in our benchmark fail to generalize when tested on high frequency sounds (see Supplementary Figure 1).

Moreover, many model organisms (rodents, birds, and bats) are experimentally monitored in laboratory environments made of rigid and reverberant materials. The use of these materials is necessary to prevent animals from escaping experimental arenas, which is of particular concern when doing

longitudinal semi-natural experiments. For example, in attempts to mitigate reverberance using specialized equipment such as anechoic foam and acoustically transparent mesh, we found that gerbils will climb or chew through material after a short time in the arena. Therefore, use of hard plastic materials, even at the expense of being more reverberant, is required. Thus, the prevalence and character of sound reflections is a unique feature of the VCL benchmark. For variety, we also include benchmark data from an environment with sound absorbent wall material (E3).

## 2.2 Classical work on SSL in engineering and neuroscience

Conventional methods for SSL from acoustic signal processing are summarized in [12]. These methods primarily use differences in arrival times or signal phase across microphones to estimate sources; differences in volume levels are often ignored as a source of information (but see [3]). We use the Mouse Ultrasonic Source Estimation (MUSE) tool [43, 66] as a representative stand-in for these classic approaches in our benchmark experiments. An alternative method based on arrival times was recently proposed by Sterling, Teunisse, and Englitz [57] (see also [44]).

Neural circuit mechanisms of SSL have been extensively studied in model organisms like barn owls, which utilize exquisite SSL capabilities to hunt prey [31]. Neurons in the early auditory system represent both interaural timing and level differences in multiple animal species [9, 4, 7]. Behavioral studies in humans also establish the importance of both interaural timing and level differences [5], and the relative importance of these cues depends on sound frequency and the level of sound reverberation, among other factors [35, 29, 16]. Altogether, the neuroscience and psychophysics literature establishes that animals are adept at localizing sounds in reverberant environments. Moreover, in contrast to many classical SSL algorithms that leverage phase differences across audio waveforms, humans and animals use a complex combination of acoustical cues to localize sounds.

## 2.3 Deep learning approaches to SSL

SSL algorithms account for a variety of event-specific factors including sound frequency, volume, and reverberation. It is challenging to rationally engineer an algorithm to account for all of these factors and the acoustic machine learning community has therefore increasingly turned to deep neural networks (DNNs) to perform SSL. Grumiaux et al. [22] provide a recent and comprehensive review of this literature, including popular architectures, datasets, and simulation methods. Existing approaches to applying DNNs to SSL leverage a variety of input featurizations, like time-frequency representations (spectrograms) of the input audio. In our experiments, we use raw audio waveforms and DNNs with 1D convolutional layers, which are a reasonable standard for benchmarking purposes (see e.g. [63]). Similar to the existing SSL benchmarks listed above, the vast majority of published DNN models have focused on large home or office environments, which differ substantially from our applications of interest.

## 2.4 Acoustic simulations

Across a variety of machine learning tasks, DNNs tend to require large amounts of training data [27]. This is problematic, since it is labor intensive to collect ground truth localization data and curate the result to ensure accurate labels. To overcome this limitation, there is recent interest in leveraging acoustic simulations to supplement DNN training sets. Geometric acoustic simulations such as the image source method (ISM) [2] are popular, due to their relatively low computational cost, as well as their ability to preserve spatial information necessary for SSL [8][32]. Recent work has shown that use of room simulations generated using the ISM can also benefit model performance on real-world data [28] and can improve robustness by simulating a wider range of acoustic conditions than is present in an existing training dataset [49], despite perceptual limitations of the ISM. Given these trends in the field, our dataset release includes simulated environments and code for performing ISM simulations.

## 3 The VCL Dataset

The VCL Dataset consists of raw multi-channel audio and image data from 767,295 sound events with ground truth 2D position of the sound event source established by an overhead camera. We recorded synchronized audio (125 or 250 kHz sampling rate) and video (30 Hz or 150 Hz sampling rate) during

| Name | # Samples |
|------|-----------|
| **Speaker-4M-E1** | 70,914 |
| **Edison-4M-E1** | 266,877 |
| GerbilEarbud-4M-E1 | 7,698 |
| **SoloGerbil-4M-E1** | 61,513 |
| **DyadGerbil-4M-E1** | 653 |
| **Hexapod-8M-E2** | 156,900 |
| **MouseEarbud-24M-E3** | 200,000 |
| SoloMouse-24M-E3 | 549 |
| **DyadMouse-24M-E3** | 2,191 |

Table 1: Summary of datasets. Datasets in **blue** were used as training sets and for test sets when benchmarking SSL. Datasets in **red** were used as test sets when benchmarking sound attribution.

| Name | # Mics | Dimensions (m) |
|------|--------|----------------|
| E1 | 4 | Top: 0.61595 x 0.41275
Bottom: 0.5588 x 0.3556
Height: 0.3683 |
| E2 | 8 | Top: 1.2182 x 0.9144
Bottom: 1.2182 x 0.9144
Height: 0.6096 |
| E3 | 24 | Top: 0.615 x 0.615
Bottom: 0.615 x 0.615
Height: 0.425 |

Table 2: Summary of environments. The final two characters in each dataset name (refer to Table 1) specifies the environment in which it was collected.

sound generating events from point sources emanating from either a speaker or real rodents. Sound events were sampled across three environments of varying size, microphone array geometries, and building material (Figure 1A-B, Table 1-2). Ground truth positions were extracted from the video stream using SLEAP [47] or OpenCV, and vocal events from real rodents were segmented from the audio stream using DAS [55]. To assess the quality of the machine-generated ground truth labels, we sampled 50 random vocal events from each training dataset and had four researchers manually label the ground truth location in each associated video frame (Supplementary Table 2). Timestamps from sound events using speaker playback were either recorded by a National Instruments data acquisition device or pre-computed and used to generate a wav file with known sound event onset times.

Brief descriptions of each dataset are included below and a more detailed description is provided in the supplemental datasheet (see "Collection Process" section). For datasets that involved speaker playback, we primarily used rodent vocalizations as stimuli (Figure 1C). In addition, we played sine sweeps in each environment which were used to compute a room impulse response (RIR, see Section 3.5). All procedures related to the maintenance and use animals were approved by the University Animal Welfare Committee at New York University and Princeton University. All experiments were performed in accordance with the relevant guidelines and regulations.

## 3.1 Speaker Datasets

The Speaker Dataset (Speaker-4M-E1) was generated by repeatedly presenting five characteristic gerbil vocal calls and a white noise stimulus at three volume levels (18 total stimulus classes) through an overheard Fountek NeoCd1.0 1.5" Ribbon Tweeter speaker. Between every set of presentations, the speaker was manually shifted two centimeters to trace a grid of roughly 400 points along the cage floor. This procedure yielded a dataset of 70,914 presentations spanning the 18 stimulus classes. Gerbil vocalizations can range in frequency from approximately 0.5-60 kHz and different vocalizations correspond to different types of social interactions in nature [60]. In this study, we selected a diverse set of commonly used vocal types vary in frequency range and ethologcial meaning.

## 3.2 Robot Datasets

The generation of the Speaker Dataset was quite labor intensive due to manual movement of the speaker, therefore the procedure was impractical for generating additional training datasets at numerical and spatial scale. To get around this issue, we developed two robotic approaches for autonomous playback of sound events. The Edison and Hexapod Datasets (Edison-4M-E1, Hexapod-8M-E2) were generated by periodically playing vocalizations through miniature speakers affixed to the robots as they performed a pseudo-random walk around the environment. The vocalizations used were sampled from a longitudinal recording of gerbil families [48].

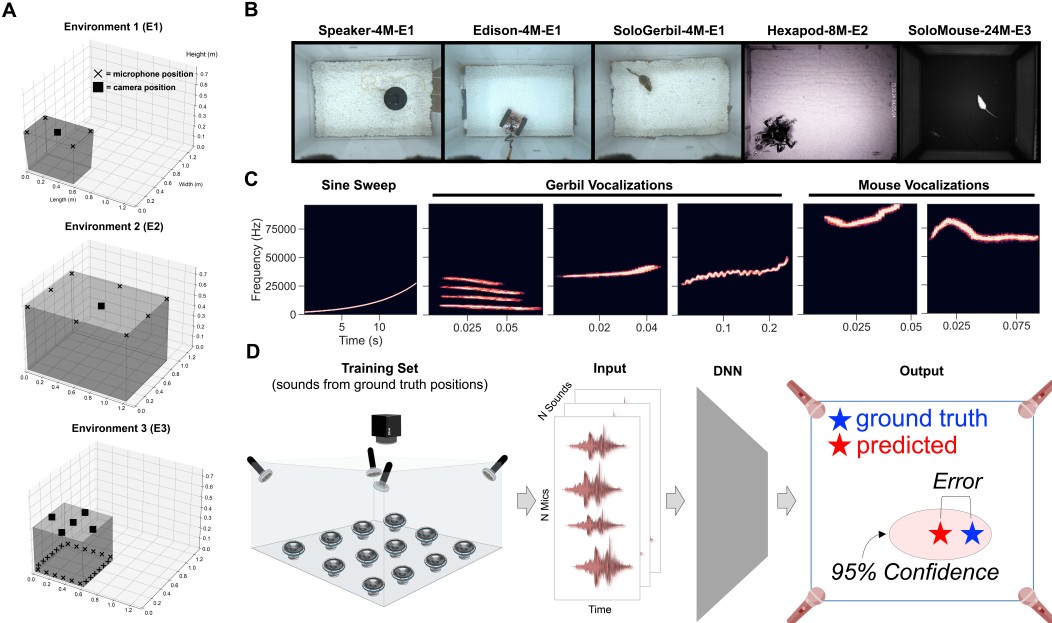

Figure 1: Overview of VCL benchmark. (A) Schematics of three laboratory arenas summarized in Table 2 showing relative size and positions of mics (X's) and cameras (squares). (B) Top-down views of different environments and training data generation modalities. (C) Examples of stimuli used for playback from Speaker, Edison, Earbud, and Hexapod datasets. (D) Schematic of pipeline depicting inputs (raw audio) and outputs (95% confidence interval).

### 3.3 Earbud Datasets

Speaker and robotic playback of vocalizations may not accurately represent the spatial usage and direction of vocalizations in real animals. To address this, we acquired two "Earbud" datasets (GerbilEarbud-4M-E1, MouseEarbud-24M-E3), in which gerbils or mice freely explored their environment with an earbud surgically affixed to their skull. We then played species typical vocalizations out of the earbud while animals exhibited a range of natural behaviors.

### 3.4 Solo/Dyad Gerbil & Mouse Datasets

Although isolated animals usually do not vocalize, we found that adolescent gerbils produce antiphonal responses to conspecific vocalizations played through a speaker. We leveraged this behavior to generate a large scale dataset, SoloGerbil-4M-E1, containing real gerbil-generated vocalizations in isolation. In addition, we elicited solo vocalizations in male mice (SoloMouse-24M-E3) by allowing female mice in estrus to explore the environment prior to male exploration.

Our ultimate goal is to use sound source estimates to attribute vocalizations to individuals in a group of socially interacting animals. To this end, we acquired vocalizations from pairs of interacting gerbils and mice (DyadGerbil-4M-E1, DyadMouse-24M-E3). Although we are unable to determine the ground truth position of vocalizations recorded from these interactions, we do know the locations of both potential sources and can therefore ascertain whether our model generates predictions with zero, one, or two animals within its confidence interval (See Task 2 below).

### 3.5 Synthetic Datasets

Since DNNs often require large training datasets and generation of datasets in the domain of SSL is laborious, we explored the use of acoustic simulations for supplementing real training data (Figure 2). We generated *in silico* models of our three environments accounting for physical measurements of the geometry, microphone placement, microphone directivity, and estimates of the material absorption coefficients (calculated via the inverse Sabine formula on room impulse response measurements with a sine sweep excitation). Code to reproduce these simulations and adapt them to new environments is

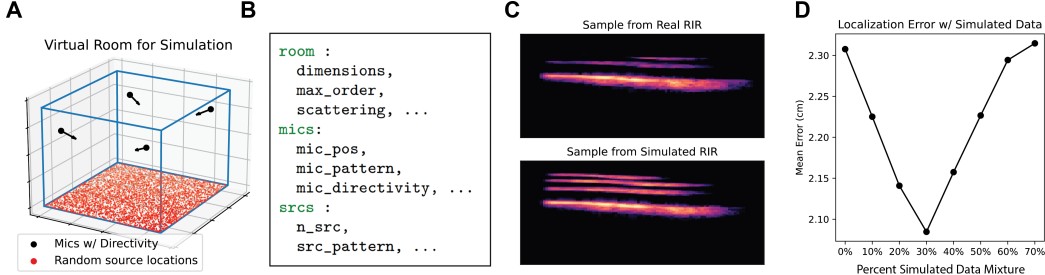

Figure 2: (A) Visualization of virtual room used for sythetic RIR generation via ISM (B) Sample of a room configuration YAML used to specify room geometry for simulations (C) Spectrograms comparing vocalizations convolved with recorded RIRs and simulated RIRs (D) Localization error as a function of added simulated data to the training corpus.

included in our code package accompanying the VCL benchmark. In preliminary experiments, we found that training DNNs on mixtures of real and simulated data can benefit performance (Figure 2D). DNNs trained exclusively on simulated data and evaluated on real data yields performance that marginally exceeds chance, but fails to match up to DNNs trained on smaller real datasets. This gap in performance indicates that our virtual acoustic models do not adequately simulate real acoustic environments. We believe that future work incorporating more robust acoustic simulations can bridge this gap. For these reasons, we do not include simulated data in benchmark experiments described below.

## 4    Benchmarks on VCL

We established a benchmark on the VCL Dataset using two distinct tasks.

- **Task 1 - Sound Source Localization:** Compare the performance of classical sound source localization algorithms with deep neural networks.

- **Task 2 - Vocalization Attribution:** Assign vocalizations to individuals in a dyad.

We evaluated performance on Task 1 using datasets with a single sound source (marked in **blue** in Table 1). We calculated the centimeter error between ground truth and predicted positions. Our aim is to achieve errors less than or equal to ~1 cm, as this is the approximate resolution required to attribute sound events to individual animals.[1] We also sought to benchmark the accuracy of model-derived confidence intervals. That is, for each prediction the model should produce a 2D set that contains the sound source with specified confidence (e.g. a 95% confidence set fail to contain the true sound source on only 5% of test set examples). Following procedures from Guo et al. [25], we plot reliability diagrams and report the expected calibration error (ECE) and maximal calibration error (MCE).

We evaluated performance on Task 2 using datasets with two potential sound sources (marked in **red** in Table 1). For Task 2, we report the number of animals inside the 95% confidence set of model predictions. For each sound event, the model can predict zero, one, or two animals within its confidence set. We report the frequency of each of these outcomes and interpret them as follows. First, if only one animal is within the confidence set, the model attributes the vocalization to that animal. We cannot for verify whether this attribution is correct because (unlike the datasets used in Task 1) we do not have ground truth measurements of the sound source. Second, if two animals are within the confidence set, then the model is unable to reliably attribute the sound to an individual. This outcome is neither correct nor incorrect. Finally, if zero animals are within the confidence set, then the model has falsely attributed the sound to a region. This outcome is clearly incorrect and should ideally happen less than 5% of the time when using a 95% confidence set.

---

[1]See, for example, Figure 1D in [57] for a distribution of inter-animal distances during natural social behavior.

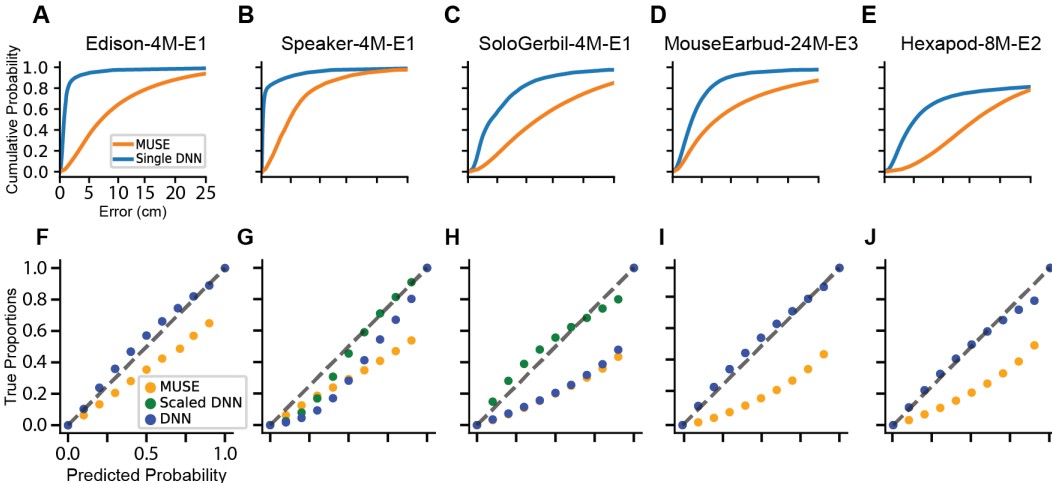

Figure 3: Benchmark performance. (A-E) Cumulative error distributions for MUSE and neural networks. (F-J) Reliability diagrams for MUSE (orange) and neural networks with (green) and without (blue) temperature scaling on heldout data from each dataset.

## 4.1 Convolutional Deep Neural Network

The network consists of 1D convolutional blocks connected in series. The network takes in raw multi-channel audio waveforms and outputs the mean and covariance of a 2D Gaussian distribution over the environment. Intuitively, the mean represents the network's best point estimate of the sound source and the scale and shape of the covariance matrix corresponds to an estimate of uncertainty. The network is trained with respect to labeled 2D sound source positions to minimize a negative log likelihood criterion—this is a proper scoring rule [18] which encourages the model to accurately portray its confidence in the predicted covariance. That is, the 95% upper level set of the Gaussian density should ideally act as a 95% confidence set. However, in line with previous reports, we sometimes observe that DNN confidence intervals are overconfident. In these cases, we use a temperature scaling procedure to calibrate the confidence intervals [25]. Further details on data preprocessing, model architecture, training procedure are provided in the Supplement.

## 4.2 MUSE Baseline Model

We compare the DNNs to a delay-and-sum beamforming approach used by neuroscientists called MUSE [43, 66]. MUSE works by computing cross-correlation signal between all pairs of microphone signals across hypothesized sound source locations, using the distance between microphones and the speed of sound to compute arrival time delays. The location that maximizes the summed response power over all microphones is then selected as a point estimate. We generate 95% confidence sets using a jackknife resampling technique proposed in Warren, Sangiamo, and Neunuebel [66].

## 4.3 Task 1 Results

Deep neural networks consistently produced estimates closer to the ground truth source than MUSE (Figure 3 A-E, Table 3). DNN performance was particularly strong on the Edison-4M-E1 and Speaker-4M-E1 datasets, achieving <1 cm error on 80.6% and 66.0% on the respective test sets. As mentioned above, this level of resolution should enable attribution of most vocalizations in realistic social encounters in rodents [57]. DNNs also outperformed MUSE on the remaining three datasets; however, they achieved sub-centimeter errors on less than 10% of the test set in all cases.

Moreover, we found that DNNs provide more accurate estimates of uncertainty relative to MUSE, as calculated by ECE and MCE (Table 4). This performance difference is visible in reliability diagrams, which show that MUSE predictions are over-confident (Figure 3F-J).

| Dataset | DNN Error (cm) | | | MUSE Error (cm) | | |
|---|---|---|---|---|---|---|
| | Mean | Median | % <1cm | Mean | Median | % <1cm |
| Speaker-4M-E1 | 1.4 | 0.2 | 80.6% | 6.4 | 4.8 | 5.9% |
| Edison-4M-E1 | 1.4 | 0.7 | 66.0% | 9.5 | 7.1 | 3.1% |
| SoloGerbil-4M-E1 | 6.1 | 4.0 | 5.2% | 14.3 | 11.9 | 0.9% |
| Hexapod-8M-E2 | 12.9 | 5.2 | 4.8% | 18.1 | 15.6 | 0.3% |
| MouseEarbud-24M-E3 | 4.1 | 2.6 | 8.7% | 11.3 | 7.6 | 3.3% |

Table 3: Summary of sound source localization errors for Task 1.

| Dataset | DNN | | Scaled DNN | | MUSE | |
|---|---|---|---|---|---|---|
| | ECE | MCE | ECE | MCE | ECE | MCE |
| Speaker-4M-E1 | 0.13 | 0.23 | 0.05 | 0.13 | 0.17 | 0.36 |
| Edison-4M-E1 | 0.03 | 0.07 | - | - | 0.12 | 0.25 |
| SoloGerbil-4M-E1 | 0.22 | 0.42 | 0.05 | 0.11 | 0.24 | 0.47 |
| Hexapod-8M-E2 | 0.03 | 0.11 | - | - | 0.22 | 0.40 |
| MouseEarbud-24M-E3 | 0.03 | 0.06 | - | - | 0.25 | 0.45 |

Table 4: Expected Calibration Error (ECE) and Maximum Calibration Error (MCE) for Task 1.

## 4.4 Task 2 Results

To test the ability of our DNNs to assign vocalizations to individuals in dyadic interactions, we used DNNs trained on single-agent datasets, MouseEarbud-24M-E3 and SoloGerbil-4M-E1 respectively, to compute confidence bounds on vocalizations from the dyadic datasets MouseDyad-24M-E3 and GerbilDyad-4M-E1. As described above, we used temperature rescaling to ensure DNN confidence sets were well-calibrated. While we were capable of assigning between 19-29% of these calls to a single animal, over half of the vocalizations in each interaction yielded a confidence bound containing both animals (Table 5). Methods to resolve these shortcomings remain a focus of future work.

## 5 Limitations

Neuroscientists are interested in localizing sounds across a broad range of settings. We aimed to cover multiple rodent species (gerbils and mice), environment sizes, and microphone array geometries in this initial release. We also leveraged robots and head-mounted earbud speakers to collect sounds with known ground truth. However, this benchmark does not yet cover all use cases in neuroscience. Other commonly used model species—e.g., marmosets[14], bats[61], and various bird species[6]—are of great interest and are not covered by the current benchmark. Our experiments show that deep neural networks trained to localize sounds can fail to generalize across vocal call types (see Supplementary Figure 1). It would therefore be valuable to expand this benchmark to include a wider variety of animal species, call types, and increase the number of training samples. To this end, we include additional datasets which were not used in Task 1 due to their relatively small size (GerbilEarbud-4M-E1, SoloMouse-24M-E3), which will aid future experiments assessing generalization performance across datasets (e.g. train on Speaker-4M-E1, predict on GerbilEarbud-4M-E1).

Our current benchmark only provides images from a single camera view, which can be used to localize sounds in 2D. While this agrees with current practices within the field [43, 56, 39] and is in line with the equipment readily available to most labs, it is insufficient to infer 3D body pose information. One could imagine that knowing the 3D position and 3D heading direction of a vocalizing rodent could provide a more rich and effective supervision signal to train a deep network. A number of 3D pose tracking tools for animal models have been developed in very recent years [68, 42, 30, 37, 13]. These tools could be leveraged if future benchmarks collect multiple camera views. Ultimately, it would be

| | Gerbil Dyad | | | Mouse Dyad | | |
|---|---|---|---|---|---|---|
| # Animals Captured | 0 | 1 | 2 | 0 | 1 | 2 |
| Percentage | 6.1% | 28.6% | 65.2% | 8.9% | 19.4% | 71.7% |

Table 5: Vocalization attribution results. Number of animals captured within the 95% confidence set.

useful to compare performance across 3D and 2D benchmarks, to ascertain whether the sound source localization problem is indeed easier in one setting or the other.

# 6 Discussion

SSL is a well-known and challenging problem. We collected a variety of datasets and developed benchmarks to assess these challenges in the context of neuroethological experiments in vocalizing rodents. This involves localizing sounds in reverberant environments across a very broad frequency range (including ultrasonic events), distinguishing our work from more standard SSL benchmarks and algorithms. Our experiments reveal that DNNs are a promising approach. In controlled settings (Edison-4M-E1 and Speaker-4M-E1 datasets), DNNs achieved sub-centimeter resolution. In larger environments (Hexapod-8M-E2) and in datasets with uncontrolled 3D variation in sound emissions (SoloGerbil-4M-E1 and MouseEarbud-24M-E3), DNN performance was less impressive, but still outperformed a well-established benchmark algorithm (MUSE), that is currently utilized.

In addition to continuing to experiment with advances in machine vision/audio, we are also interested in exploring performance improvements due to hardware optimization. Parameters such as number of microphones, their positions/directivity, and environment reverberance can all affect SSL performance. Future experiments will leverage acoustic simulations to explore this parameter space. Initial results suggest that varying the amount of reverberation in an environment drastically affects SSL performance and that this effect is more pronounced in MUSE than DNNs (see Supplementary Figure 3). Moreover, we assessed whether specific acoustic or environmental features within the dataset affect model performance (Supplementary Figure 4). Sound power and distance from center of environment have a compelling effect on performance, where low power sounds and sounds that occur far away from the center of the arena (i.e. close to the walls) are difficult to localize. Fundamental frequency does not have a strong relationship to performance.

The ultimate goal of most neuroscientists in this context is to attribute vocal calls to individuals amongst an interacting social group. Accurate SSL would enable this, but it is also possible to reframe this problem as a direct prediction task. Specifically, given a video and audio recording of $K$ interacting animals with ground truth labels for the source of each sound event, DNNs could be trained to perform $K$-way classification to identify the source. Future work should investigate this promising alternative approach, as it would enable DNNs to jointly leverage information from audio and video data as network inputs. On the other hand, we note several challenges that must be overcome. First, establishing ground truth in multi-animal recordings is non-trivial, though feasible in certain experiments [17, 50, 64]. Second, DNNs trained to process raw video can have trouble generalizing across recording sessions due to subtle changes in lighting or animal appearance [67, 53]. Finally, we note that at least $K = 2$ animals are required to make the problem nontrivial (when $K = 1$ the DNN could ignore the audio input to predict the source). It will be important to establish a flexible DNN architecture that can make accurate predictions even when the animal group size, $K$, is altered (see e.g. [70]). It is already possible to use the VCL datasets to explore these possibilities. For example, one could use audio and video data taken from the same or different sound events to train a DNN with a multimodal contrastive learning objective (see e.g. [59], for a related concept).

In summary, there are many promising, but under-investigated, machine learning methodologies for annotating vocal communication in rodents. The VCL benchmark is our attempt to spark a broader community effort to investigate the potential of these computational approaches. Indeed, collecting and curating these datasets is labor-intensive and in our case involved collaboration across multiple neuroscience labs. To our knowledge, very little (if any) comparable data containing raw audio and video from many thousands of rodent vocal calls currently exists in the public domain. Thus, we expect the VCL benchmark will enable new avenues of research within computational neuroscience.

**Acknowledgements and Ethics Statement**

We do not foresee any negative societal impacts arising from this work. We thank Megan Kirchgessner (NYU), Robert Froemke (NYU), and Marcelo Magnasco (Rockefeller) for discussions and suggestions regarding SSL applications in neuroscience. This work was supported by the National Institutes of Health R34-DA059513 (AHW, DHS, DMS), National Institutes of Health R01-DC020279 (DHS), National Institutes of Health 1R01-DC018802 (DMS, REP), National Institutes of Health Training Program in Computational Neuroscience T90DA059110 (REP), New York Stem Cell Foun-

dation (DMS), CV Starr Fellowship (BM), EMBO Postdoctoral Fellowship (BM), National Science Foundation Award 1922658 (CI).

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
