# 1 Supplementary Information

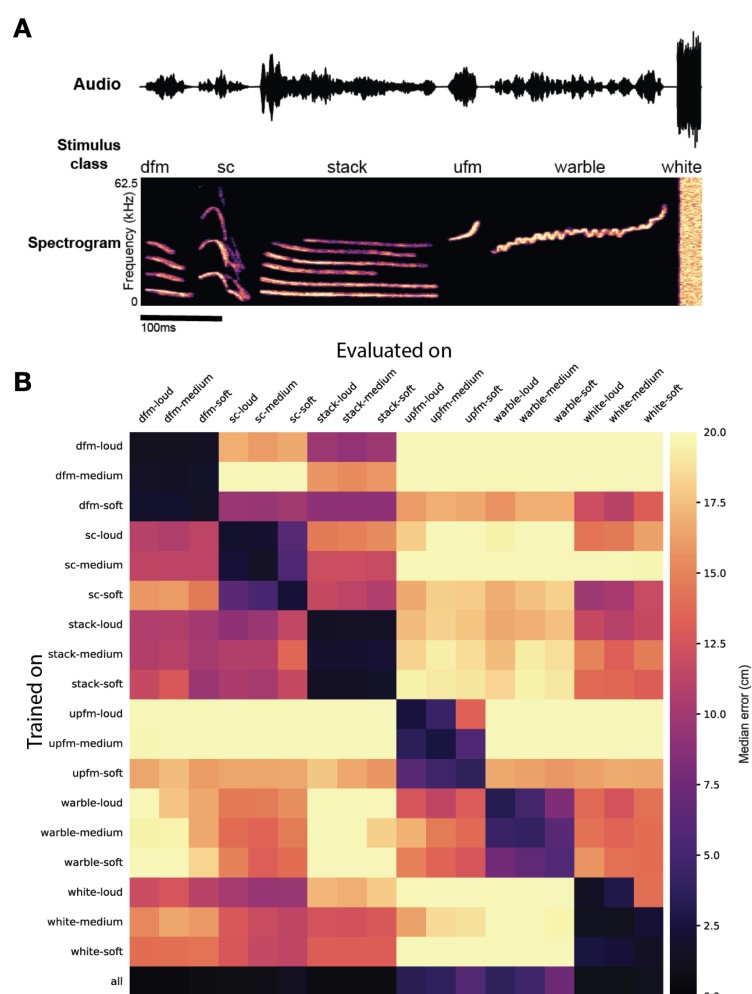

Figure 1: Generalizability across stimulus types. A.) Performance of models trained on single stimuli from Speaker-4M-E1 dataset and evaluated on all other stimulus types. (B) Stimuli used for speaker data set (dfm = down frequency modulated, sc = soft chirp, stack = harmonic stack, ufm = up frequency modulated)

Ultimately, we aim to create a tool that can be easily adapted by other labs which may have different recording environments. Additionally, we wish to utilize the tool for long-term recordings in which the types of vocalizations encountered may change over time as the animals enter new stages of life. As such, we have significant interest in the model's ability to generalize to unfamiliar vocal calls

To explore this, we tested the ability of deep networks to generalize to new vocal calls with different acoustic features. We partitioned the Speaker-4M-E1 Dataset according to stimulus type (Supplementary Figure 2A), trained a deep neural network on each subset, and measured its performance on every stimulus type individually (Supplementary Figure 2B). We found that while many models could generalize to new stimuli with performance exceeding chance, their ability to do so is greatly overshadowed by their performance on their own subsets. Models trained on a single stimulus type generalized well to the same stimulus at different volumes. (Supplementary Figure 2B, 3x3 block structure). This suggests that the networks are adapted to the statistics of the training set, and that training on a range of vocalizations with diverse spectral features will be necessary to achieve good performance across experimental cohorts, each of which may utilize slightly different vocal calls.

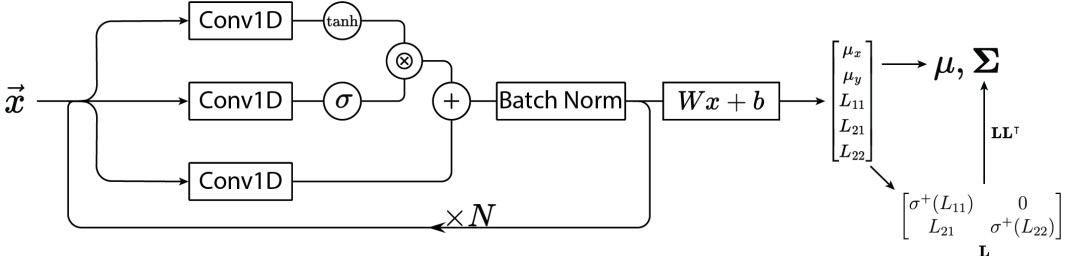

Figure 2: Network architecture.

| Layer | Channels | Downsample |
|---|---|---|
| 1 | 32 | No |
| 2 | 32 | Yes |
| 3 | 64 | No |
| 4 | 64 | Yes |
| 5 | 128 | No |
| 6 | 128 | Yes |
| 7 | 256 | No |
| 8 | 256 | Yes |
| 9 | 512 | No |
| 10 | 512 | Yes |

Table 1: Model Architecture Hyperparameters. Our model consists of 10 convolutional blocks. All use a kernel size of 33, dilation of 1, and stride of 1.

Mirroring gated linear units [11] and WaveNet [62], we apply tanh and sigmoid nonlinearities to the output of convolutions and multiply them element-wise. We add this product to the result of a third convolution and apply batch normalization to the sum. On layers with temporal downsampling, we perform average pooling with a stride and kernel size of 2 prior to normalization. On our datasets with four microphones, we incorporate pairwise cross-correlations of the microphone signals by concatenating the central elements of each cross-correlogram into a vector, passing it through a shallow MLP, and concatenating the result to the output of the final convolutional block. The model outputs the mean and covariance of a 2D Gaussian distribution with covariance specified by a Cholesky factor matrix. To parametrize the 2D gaussian posterior distribution, we first average the output of the final convolutional block over its time dimension and linearly project it to five components. Two of these determine the distribution's mean and the other three parametrize the Cholesky decomposition of the distribution's covariance matrix. In order to ensure the Cholesky factor has positive diagonals, we apply the softplus nonlinearity to the diagonal elements. During training, we evaluate the log likelihood of the ground truth positions with respect to the 2D Gaussians output by the network. We minimize the negative log likelihood using stochastic gradient descent with momentum. Throughout 50 epochs, we anneal the learning rate to 0 using a cosine schedule. We do not use weight decay.

For data preprocessing, we normalize the audio by ensuring a zero mean and unit variance across all elements, rather than scaling each channel individually. This approach ensures amplitude differences between channels are preserved after normalization. Throughout training, we apply various augmentations to the audio to enhance sample efficiency and performance on the validation set. As vocalization lengths vary substantially, we randomly crop them to a standardized length of 8192 samples (65.5ms at 125kHz) to facilitate batched computations. Additional augmentations include temporal masking, the introduction of white noise, and phase inversion. With the exception of cropping, which is applied universally to all samples, each augmentation has a 50% chance of being applied to a given vocalization.

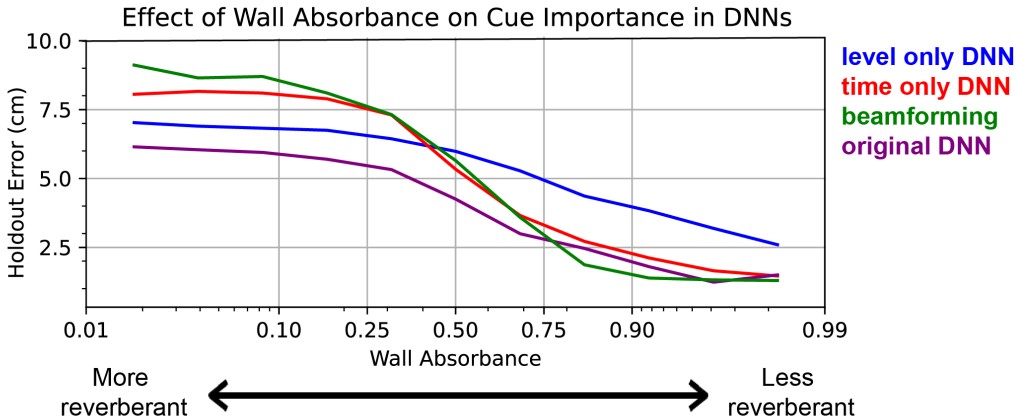

Figure 3: SSL performance with varying environmental reverberance.

We explored whether SSL performance systematically varied as a function of reverberance using acoustic simulations. First, we simulated an E1 environment, then simulated microphone signals from 50,000 gerbil vocalizations randomly sampled from [48]. Next, we compared DNN vs. MUSE (beamforming) performance and showed that DNNs (purple) outperform MUSE (green) in reverberant conditions and achieve equal performance in non-reverberant conditions. Furthermore, we explored which cues (temporal or level, i.e. akin to ITD and ILD cues used by animals) DNNs relied on for SSL. We created augmented training sets that either scrambled level differences between microphone channels (thereby only maintaining reliable time differences, red) or scrambled time differences (thereby only maintaining reliable level differences, blue). We find that time-only DNN performance matches MUSE, which is consistent with the fact that MUSE and other beamforming algorithms are time-only models. In addition, we find that level-only models outperform time-only models in reverberant conditions, but do worse in non-reverberant enrionments. Intriguingly, DNNs trained with both time and level (purple) perform better than level-only models in reverberant environments, suggesting that DNNs are making use of both available cues, though likely relying more on level. Future studies will aim to better understand how DNNs and biological neural networks balance the relative use of these two cues in reverberant listening conditions.

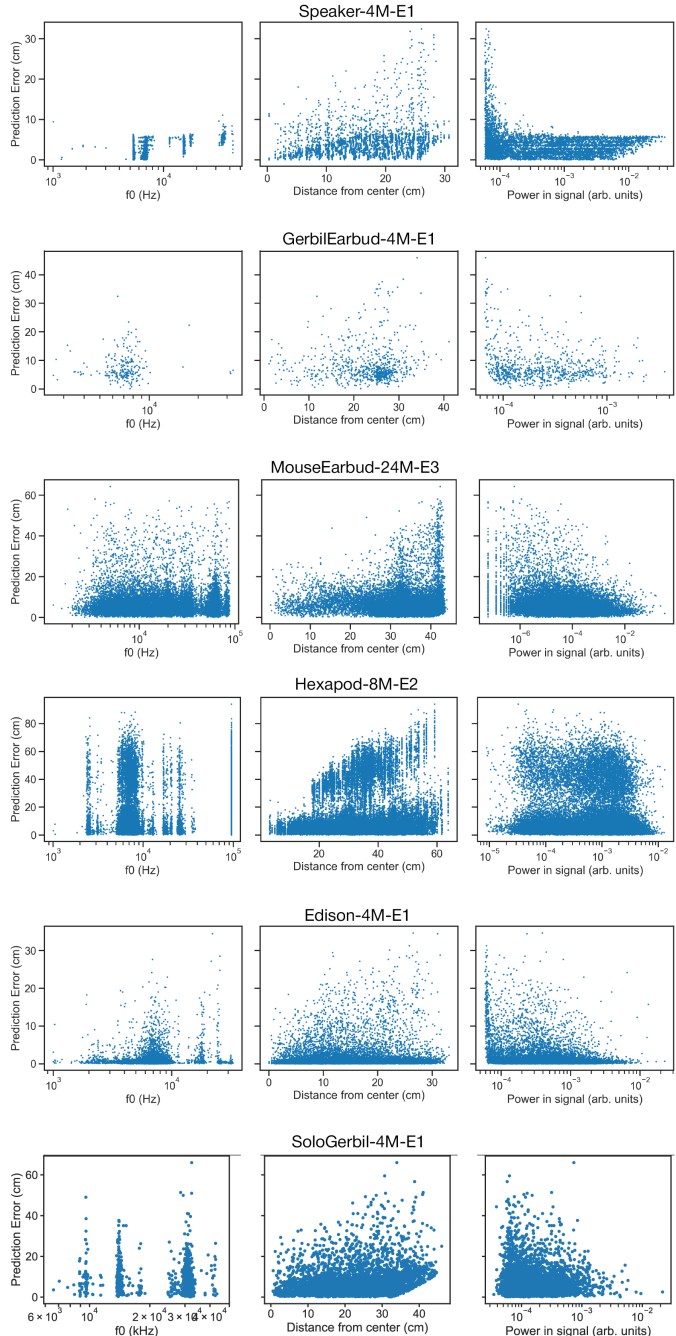

Figure 4: Effect of acoustic and environmental factors on localization performance.

To assess whether variation in localization performance relates to interpretable features in the dataset, we plotted the fundamental frequency, power, and distance to center of each sample in the test set as a function of localization error. Indeed, samples that are lower power and further from the center (i.e. next to the wall) are more difficult to localize. There is not an appreciable relationship between frequency of sample and localization error.

| Dataset | Error (px) | | | | |
|---|---|---|---|---|---|
| | Mean | Median | Max | Min | Human std |
| Speaker-4M-E1 | 5.8 | 6.2 | 12.3 | 0.8 | 0.6 |
| Edison-4M-E1 | 5.4 | 5.9 | 12.8 | 0.9 | 1.3 |
| SoloGerbil-4M-E1 | 8.4 | 6.5 | 38.8 | 0.4 | 1.0 |
| Hexapod-8M-E2 | 7.2 | 6.7 | 16.3 | 1.0 | 2.6 |
| MouseEarbud-24M-E3 | 4.9 | 4.2 | 23.9 | 0.7 | 1.5 |

Table 2: Analysis of error in machine-labeled ground truth.

Four researchers were tasked with annotating ground truth locations of the sound source within 50 video frames from each training dataset. We compared these human ground truth annotations with machine labeled ground truth locations used for SSL model training in this benchmark. The error in the machine label for each image was computed as the pixel distance between that label and the centroid of the human labels for the image in pixel space. We report the mean, median, maximum, and minimum error for each training dataset in addition to the average amount of deviation from the centroid in the human labels. SoloGerbil-4M-E1 exhibited a higher than expected error in machine-labeled ground truth locations, which at least partially explains the relatively high sound localization error for this dataset (Figure 3C, Table 3). Future releases of this benchmark will improve ground truth labels.

# 2 Datasheet for VCL

## Motivation For Dataset Creation

### 2.1 For what purpose was the dataset created? (e.g., Was there a specific task in mind? Was there a specific gap that needed to be filled?)

Communication disorders affect more than 45 million individuals in the United States [41], however the neural mechanisms that underlie these disorders are poorly understood. The field of neuroscience has developed a number of animal models to study the neural basis of vocal communication, including social rodents, who naturally vocalize during social interactions. The VCL dataset and benchmark were established to address the problem of sound-source localization (SSL) and source attribution in the context of rodent social-vocal interactions. More colloquially, in a group of vocalizing individuals: **who said what**? Existing non-invasive approaches rely on classical signal processing algorithms, though their performance is contingent upon specialized acoustical hardware which is not ideal for next-generation neuroscience experiments (see main text Section 1 and 2.1). Therefore, an off-the-shelf SSL tool which works in standard reverberant laboratory environments is needed.

Advances in deep learning for machine audio are well poised to help address these challenges [22], however there is little interaction between the fields of machine audio and neuroscience. One of the functions of this dataset and benchmark release is to facilitate collaboration between these fields by formalizing the neuroscientific problem in terms of a machine learning problem. To this end, we acquired the first ever large-scale SSL dataset with ground truth labels for SSL and source attribution in rodent social interactions. In addition, we introduce two tasks which serve as benchmarks:

- **Task 1 - Sound Source Localization:** Compare the performance of classical sound source localization algorithms with deep neural networks.
- **Task 2 - Vocalization Attribution:** Assign vocalizations to individuals in a social dyad.

### 2.2 What (other) tasks could the dataset be used for?

Recent work[28] has shown that sound localization DNNs trained on synthetic data can be used to reliably predict sound sources in held-out real-world data. Indeed, preliminary experiments (see main text, Figure 2D) suggest that this is also true for the VCL dataset. Future experiments will explore how DNNs trained on synthetic only vs. real data augmented with synthetic data (with different proportions real:synthetic) affect performance.

This dataset can also be used to assess SSL generalization. For example, Supplementary Figure 1 shows that predictions from DNNs trained on a single stimulus class do not generalize well to other stimulus classes. Future tasks will explore different architectures and input embeddings that may facilitate generalization.

### 2.3 Who created this dataset (e.g., which team, research group) and on behalf of which entity (e.g., company, institution, organization)?

The dataset was created collaboratively between the Williams (NYU), Sanes (NYU), Schneider (NYU), Falkner (Princeton), and Murthy (Princeton) labs. Datasets from environments E1/E2 were acquired at NYU, E3 at Princeton.

### 2.4 Who funded the creation dataset?

This work was supported by the National Institutes of Health R34-DA059513 (AHW, DHS, DMS), National Institutes of Health R01-DC020279 (DHS), National Institutes of Health 1R01-DC018802 (DMS, REP), National Institutes of Health Training Program in Computational Neuroscience

T90DA059110 (REP), New York Stem Cell Foundation (DMS), CV Starr Fellowship (BM), EMBO Postdoctoral Fellowship (BM), National Science Foundation Award 1922658 (CI).

## 2.5 Any other comment?

None.

**Dataset Composition**

## 2.6 What are the instances?(that is, examples; e.g., documents, images, people, countries) Are there multiple types of instances? (e.g., movies, users, ratings; people, interactions between them; nodes, edges)

A single instance in the dataset consists of two features:

- Raw multi-channel audio data recorded from a microphone array.

- Ground truth source position (XY) of audio data.

## 2.7 How many instances are there in total (of each type, if appropriate)?

767,295 total sound events (instances) from 9 unique conditions. See main text, Table 1.

## 2.8 What data does each instance consist of? "Raw" data (e.g., unprocessed text or images)? Features/attributes? Is there a label/target associated with instances? If the instances related to people, are subpopulations identified (e.g., by age, gender, etc.) and what is their distribution?

Each instance consists of raw multi-channel audio with a label indicating where in 2-dimensional space the sound came from. The sound itself comes from either speaker playback of vocalizations/sine sweeps or a real vocalizing animal.

## 2.9 Is there a label or target associated with each instance? If so, please provide a description.

Yes, the label for each instance is the 2-dimensional location of the sound source.

## 2.10 Is any information missing from individual instances? If so, please provide a description, explaining why this information is missing (e.g., because it was unavailable). This does not include intentionally removed information, but might include, e.g., redacted text.

None.

## 2.11 Are relationships between individual instances made explicit (e.g., users' movie ratings, social network links)? If so, please describe how these relationships are made explicit.

Yes, samples were acquired from distinct environments, sources, and represent different stimulus types. See main text, Table 1 for additional detail.

**2.12 Does the dataset contain all possible instances or is it a sample (not necessarily random) of instances from a larger set? If the dataset is a sample, then what is the larger set? Is the sample representative of the larger set (e.g., geographic coverage)? If so, please describe how this representativeness was validated/verified. If it is not representative of the larger set, please describe why not (e.g., to cover a more diverse range of instances, because instances were withheld or unavailable).**

The dataset contains all possible instances.

**2.13 Are there recommended data splits (e.g., training, development/validation, testing)? If so, please provide a description of these splits, explaining the rationale behind them.**

We used a random 80:10:10 train, test, validation split with a predetermined random seed. The exact splits are available for download on the dataset website.

**2.14 Are there any errors, sources of noise, or redundancies in the dataset? If so, please provide a description.**

None that we are aware of.

**2.15 Is the dataset self-contained, or does it link to or otherwise rely on external resources (e.g., websites, tweets, other datasets)? If it links to or relies on external resources, a) are there guarantees that they will exist, and remain constant, over time; b) are there official archival versions of the complete dataset (i.e., including the external resources as they existed at the time the dataset was created); c) are there any restrictions (e.g., licenses, fees) associated with any of the external resources that might apply to a future user? Please provide descriptions of all external resources and any restrictions associated with them, as well as links or other access points, as appropriate.**

It is self-contained.

## Collection Process

**2.16 What mechanisms or procedures were used to collect the data (e.g., hardware apparatus or sensor, manual human curation, software program, software API)? How were these mechanisms or procedures validated?**

**E1 Datasets**

Audio, video, and speaker playback were synchronously acquired and triggered using custom python software, as per [48]. Four ultrasonic microphones (Avisoft CM16/CMPA48AAF-5V) connected to an Avisoft preamplifier were recorded by a National Instruments data acquisition device (PCI-6143) via BNC connection with a National Instruments terminal block (BNC-2110). The recording was controlled using the NI-DAQmx library (https://github.com/ni/nidaqmx-python) which wrote samples to disk at a 125 kHz sampling rate. Video (FLIR USB Backfly S) frames were externally triggered at 30 Hz via the National Instruments device and written to disk using the FLIR Spinnaker SDK, PySpin. Pre-computed wav files were stored on a Raspberry Pi 4B and played back using the "play" command from the SoX library. Audio signals from the Raspberry Pi were sent to a digital-to-analog converter (HiFiBerry DAC2 Pro), then to an amplifier (Tucker Davis Technologies SA-1), then finally to a speaker.

**E2 Datasets**

Audio and video were synchronously acquired in a similar fashion to E1 data, with two notable exceptions. First, instead of externally triggering camera frames, we recorded timestamps generated by a camera synchronization device at 30 Hz (e3 Vision Hub + Camera, White Matter LLC). Second,

we recorded 8-channel audio using a different National Instruments configuration (PXI-6143 IO Module mounted in a PXIe-1071 chassis) at a sampling rate of 125 kHz. See **Hexapod Dataset** Section for information about audio playback in this environment.

### E3 Datasets

Similarly to E1 and E2, audio and video were synchronously acquired using custom python scripts. 24 Avisoft microphones (CM16/CMPA) were connected to an Avisoft UltraSoundGate 1216H (analog-to-digital converter and pre-amplifier) and written to disk at a sampling rate of 250 kHz. Video (FLIR USB Backfly S) frames were externally triggered at 150 Hz using a Loopbio triggerbox. Audio and video were synchronized posthoc using 1.) TTLs sent every frame from the triggerbox to the Avisoft UltraSoundGate and 2.) by aligning simultaneous pulses sent from an Arduino to the Avisoft UltraSoundGate and to trigger LEDs detected by the camera. Audio playback was performed in the same manner as E1.

### Speaker Dataset

Stimuli were played back with the system described in E1 using a Fountek NeoCD1 Ribbon Tweeter speaker. Upon playback of an audio signal, a TTL was sent from the Raspberry Pi to the National Instruments system and a timestamp for each playback was logged. The speaker was positioned facing towards the ceiling of the arena and moved in 2cm increments by hand to tile the arena floor uniformly. At each position in the arena, every stimulus class (see Supplementary Section 1.1) was played 10 times (180 stimuli total, per position). The ground truth position of the sound source was obtained using OpenCV.

### Edison Dataset

A Bose SoundTrue Ultra in-ear earbud was magnetically mounted to the top of an Edison Robot v2 (https://meetedison.com/). The robot was programmed to perform a random walk using the EdPy library (https://www.edpyapp.com/) and using the speaker playback system detailed in E1, we played randomly sampled vocalizations from gerbil families [48] out of the earbud speaker every 50ms. The ground truth position of the sound source was obtained using supervised keypoint tracking with SLEAP [47].

### Hexapod Dataset

The Hexapod dataset was generated using a Freenove Hexapod Robot Kit (https://freenove.com/). The kit features a 6 legged robot independently powered by two 18650 batteries and controlled by a custom microcontroller developed by Freenove. A HC-SR04 ultrasonic sensor is mounted at the front of the hexapod to ensure it avoids collisions with walls. Additionally, a second system is integrated into the hexapod, comprising a Raspberry Pi 4B, HifiBerry DAQ2 Pro, and a MAX9744 amplifier (Adafruit: 1752), all housed in 3D printed cases. This setup plays gerbil vocalizations through a Peerless by Tymphany tweeter speaker (DigiKey: OX20SC00-04-ND) mounted on a pan-tilt servo mechanism. This system is powered by a portable Anker power bank. The tweeter speaker is enclosed in a 3D printed box marked with multiple ArUco markers, enabling continuous tracking of the speaker's 3D position. The robot is designed to operate untethered and is controlled wirelessly.

The control loop starts with the Raspberry Pi, which sends a signal to the hexapod microcontroller to start the robot's gait. After a set number of steps, the hexapod microcontroller issues a stop signal to the Raspberry Pi and waits for another start signal before resuming movement. Meanwhile, the Raspberry Pi on receiving the stop signal transmits vocalizations through the HiFiBerry DAC and MAX9744 amp to the speaker. These vocalizations are emitted at various angles using the pan-tilt servo mechanism. The hexapod resumes movement only after completing a full cycle of speaker positions and vocalizations. A complete cycle involves four distinct pan servo positions—North, North East, East, South East, and South—and five tilt servo positions—0, 45, 90, 135, and 180 degrees. At each position, two sine frequencies, a sine sweep and 100 different vocalizations are played. The sine frequencies are used to detect the start of the audio sequence and the sine sweep is used to calibrate the room impulse response for simulations.

**Earbud Datasets**

Neodymium block magnets (K+J Magnetics: B222G-N52) were glued into a custom 3D printed housing, then affixed to the skulls of anesthetized gerbils or mice using dental cement or cyanoacrylate glue. A block magnet was then glued to an earbud (either Bose SoundTrue Ultra in-ear or Sony MDREX15LP in-ear) to allow for easy application and removal of the earbud to the animal's head. After surgical recovery, animals were allowed to freely explore E1 (gerbils) or E3 (mice) with the head-mounted earbud. For gerbils, vocalizations were played back out of the earbud in the same fashion as the Edison Dataset. This process generated the GerbilEarbud-4M-E1 Dataset. For mice, a representative collection of ultrasonic vocalization types were used to generate a wav file containing 10,000 vocalizations that were played back over a period of 20 minutes (MouseEarbud-24M-E3 Dataset). The ground truth position of the sound source was obtained using supervised keypoint tracking with SLEAP.

**Solo Gerbil Dataset**

Adolescent gerbils respond robustly to playback of an 11-syllable, 2-second long sequence of ultrasonic vocalizations (REP, unpublished observations). We leveraged this behavior to generate ground-truth data by placing single adolescent gerbils in E1 and playing back the vocalization sequence to induce vocal responses from the isolated animal. We then tracked the XY position of the animal's nose using SLEAP and extracted vocalization onset times using the supervised audio segmenter DAS [55].

**Solo Mouse Dataset**

Male mice vocalize in response to the smell of urine from sexually receptive females [26]. Prior to free exploration of E3 by a single male mouse, estrus females were first allowed explore the arena, thereby depositing their smell across the environment. Males reliably vocalized in isolation following this procedure. We then tracked the XY position of the animal's nose using SLEAP and extracted vocalization onset times using the supervised audio segmenter DAS.

**Gerbil/Mouse Dyad Datasets**

Standard laboratory rodents naturally vocalize during social interactions, therefore gerbils or mice were recorded during dyadic social interactions. We tracked the XY position of the animals noses using SLEAP and extracted vocalization onset times using DAS. Although we do not have ground truth for which animal vocalized, we do know two possible locations of the sound source, therefore this dataset can be used for Task 2.

**2.17 How was the data associated with each instance acquired? Was the data directly observable (e.g., raw text, movie ratings), reported by subjects (e.g., survey responses), or indirectly inferred/derived from other data (e.g., part-of-speech tags, model-based guesses for age or language)? If data was reported by subjects or indirectly inferred/derived from other data, was the data validated/verified? If so, please describe how.**

The data were directly observable, as detailed above. In the case of datasets which used speaker playback, the onset times of vocalizations (i.e. instances) were automatically logged using a TTL sent to the data acquisition device. In the case of real rodent vocalizations where this was not possible, we used DAS, a deep-learning based audio segmenter, to extract the onset/offset times of vocalizations. To obtain the ground truth XY position of sound sources, we used SLEAP, a supervised keypoint tracking software, to infer the position of sound sources in the arena.

SLEAP and DAS are both supervised learning pipelines, therefore need sufficient and diverse training data to function appropriately. We validated the performance of these algorithms through iterative human-in-the-loop training, where the labeler trained a model with limited training data, then evaluated performance on heldout data. If the heldout performance was insufficient, the labeler added more training examples and retrained the model. This process was repeated until the models achieved desired performance.

**2.18  If the dataset is a sample from a larger set, what was the sampling strategy (e.g., deterministic, probabilistic with specific sampling probabilities)?**

N/A

**2.19  Who was involved in the data collection process (e.g., students, crowdworkers, contractors) and how were they compensated (e.g., how much were crowdworkers paid)?**

The dataset was acquired by graduate students and postdoctoral fellows at New York University and Princeton University. They are compensated by grants that support laboratories and/or individual grant funding (see Section 2.4).

**2.20  Over what timeframe was the data collected? Does this timeframe match the creation timeframe of the data associated with the instances (e.g., recent crawl of old news articles)? If not, please describe the timeframe in which the data associated with the instances was created.**

The datasets were acquired over a one year period from April 2023-May 2024.

**2.21  Data Preprocessing**

**2.22  Was any preprocessing/cleaning/labeling of the data done (e.g., discretization or bucketing, tokenization, part-of-speech tagging, SIFT feature extraction, removal of instances, processing of missing values)? If so, please provide a description. If not, you may skip the remainder of the questions in this section.**

Instances from experiment days were inspected to ensure that they contained XY positions that accurately labelled the sound source. In the event that a sound source position was mislabeled, we manually corrected the label(s) for those days, or used OpenCV to track the sound source instead of SLEAP (e.g. Speaker-4M-E1 Dataset). Occasionally we omitted the entire instance if no sound source was visible in the camera view.

**2.23  Was the "raw" data saved in addition to the preprocessed/cleaned/labeled data (e.g., to support unanticipated future uses)? If so, please provide a link or other access point to the "raw" data.**

No, we only provided the curated dataset.

**2.24  Is the software used to preprocess/clean/label the instances available? If so, please provide a link or other access point.**

Yes, see https://github.com/neurostatslab/vocalocator.

**2.25  Does this dataset collection/processing procedure achieve the motivation for creating the dataset stated in the first section of this datasheet? If not, what are the limitations?**

Yes.

**2.26  Any other comments**

None.

# Dataset Distribution

**2.27 How will the dataset be distributed? (e.g., tarball on website, API, GitHub; does the data have a DOI and is it archived redundantly?)**

**Project page:** `https://vclbenchmark.flatironinstitute.org`
**Code:** `https://github.com/neurostatslab/vocalocator`
The dataset DOI is: 10.5281/zenodo.11584391

**2.28 When will the dataset be released/first distributed? What license (if any) is it distributed under?**

As of this submission, the dataset is currently publicly available and distributed under a CC BY 4.0 license. The authors bear all responsibility in case of violation of rights.

**2.29 Are there any copyrights on the data?**

None.

**2.30 Are there any fees or access/export restrictions?**

None.

**2.31 Any other comments?**

None.

## Dataset Maintenance

**2.32 Who is supporting/hosting/maintaining the dataset?**

The Flatiron Institute Center for Computational Neuroscience and Scientific Computing Core.

**2.33 Will the dataset be updated? If so, how often and by whom?**

Yes, it will be updated as the neuroscience and ML community begin to use the datasets. Neuroscience labs will contribute their own datasets.

**2.34 How will updates be communicated? (e.g., mailing list, GitHub)**

GitHub + mailing list.

**2.35 If the dataset becomes obsolete how will this be communicated?**

Mailing list.

**2.36    Is there a repository to link to any/all papers/systems that use this dataset?**

N/A

**2.37    If others want to extend/augment/build on this dataset, is there a mechanism for them to do so? If so, is there a process for tracking/assessing the quality of those contributions. What is the process for communicating/distributing these contributions to users?**

A clear set of instructions for how to contribute a dataset will be included on the website. Contributors will be credited on the website/GitHub repo.



**Legal and Ethical Considerations**



**2.38    Were any ethical review processes conducted (e.g., by an institutional review board)? If so, please provide a description of these review processes, including the outcomes, as well as a link or other access point to any supporting documentation.**

Yes, text from section Section 3: All procedures related to the maintenance and use of animals were approved by the Institutional Animal Care and Use Committee (IACUC) at New York University and Princeton University. All experiments were performed in accordance with the relevant guidelines and regulations.

**2.39    Does the dataset contain data that might be considered confidential (e.g., data that is protected by legal privilege or by doctorpatient confidentiality, data that includes the content of individuals non-public communications)? If so, please provide a description.**

No.

**2.40    Does the dataset contain data that, if viewed directly, might be offensive, insulting, threatening, or might otherwise cause anxiety? If so, please describe why**

No.

**2.41    Does the dataset relate to people? If not, you may skip the remaining questions in this section.**

N/A

**2.42    Does the dataset identify any subpopulations (e.g., by age, gender)? If so, please describe how these subpopulations are identified and provide a description of their respective distributions within the dataset.**

N/A

**2.43    Is it possible to identify individuals (i.e., one or more natural persons), either directly or indirectly (i.e., in combination with other data) from the dataset? If so, please describe how.**

N/A

**2.44** **Does the dataset contain data that might be considered sensitive in any way (e.g., data that reveals racial or ethnic origins, sexual orientations, religious beliefs, political opinions or union memberships, or locations; financial or health data; biometric or genetic data; forms of government identification, such as social security numbers; criminal history)? If so, please provide a description.**

N/A

**2.45** **Did you collect the data from the individuals in question directly, or obtain it via third parties or other sources (e.g., websites)?**

N/A

**2.46** **Were the individuals in question notified about the data collection? If so, please describe (or show with screenshots or other information) how notice was provided, and provide a link or other access point to, or otherwise reproduce, the exact language of the notification itself.**

N/A

**2.47** **Did the individuals in question consent to the collection and use of their data? If so, please describe (or show with screenshots or other information) how consent was requested and provided, and provide a link or other access point to, or otherwise reproduce, the exact language to which the individuals consented.**

N/A

**2.48** **If consent was obtained, were the consenting individuals provided with a mechanism to revoke their consent in the future or for certain uses? If so, please provide a description, as well as a link or other access point to the mechanism (if appropriate).**

N/A

**2.49** **Has an analysis of the potential impact of the dataset and its use on data subjects (e.g., a data protection impact analysis)been conducted? If so, please provide a description of this analysis, including the outcomes, as well as a link or other access point to any supporting documentation.**

N/A

**2.50** **Any other comments?**

None.