# OpenReview forum: "Vocal Call Locator Benchmark (VCL) for localizing rodent vocalizations from multi-channel audio"
_NeurIPS.cc/2024/Datasets_and_Benchmarks_Track — NeurIPS 2024 Track Datasets and Benchmarks Poster_

### Official Review · Reviewer_XE72 · 2024-07-15

**Rating:** 6
**Confidence:** 5
**Correctness:** The dataset seems to have been constr…

**Review:**

There are multiple versions of the datasets which have been collected in realistic environments using several well motivated methods, ranging from simulation where loudspeaker is moved either manually along a predefined grid, or using a robot with pseudo-random walk. More realistic movement trajectories are obtained by a loudspkear attacher real rodents, and recording real rodent vocalizations.

All the approaches are well motivated. The description of details of each version of the dataset is somewhat slim. A significantly more detailed description and analysis of the properties of the obtained data would have been valuable (e.g. how long were the vocalizations, properties of the random walks and real trajectories, how exactly the ground turn positions were obtained, what were the species typical vocalizations etc.). Some analysis of the accuracy of ground truth locations would also have been valuable. Also the description of the baseline experiments is a bit thin, especially the description of Task 2 (vocalization attribution) would have deserved a bit better motivation and analysis.

**Strengths:**

The dataset contains several recordings and location measurement setups with varying realisticness which will nicely allow studying the effects of the measurment setup.

**Additional Feedback:**

-

**Clarity:**

Overall the paper is well written, but having more details about the data collection procedure would have been valuable, as well as having more detailed analysis of the collected data.

**Documentation:**

Many details about the data collection have not been given (see the review)

**Ethics:**

I do not see any ethical concerns

**Limitations:**

The factors related to different measurement setups have been discussed.  Negative social impacts have not been identified, but I cannot see those arising from this work.

**Opportunities For Improvement:**

The description of the data collection procedures and analysis of obtained data would have been more detailed, as well as description of the experimental setup.

**Relation To Prior Work:**

Previous studies are appropriately discussed.

**Summary And Contributions:**

The paper proposes a dataset for localizing rodent vocalizations in a laboratory environment. The datasets includes both simulated data where events played by a loudspeaker are recorded in different locations by microphones, as well as datasets consisting of real vocalizations. The paper also presents baseline localization experiments. This will be an important contribution for the scientific community.

---

> ### Author Rebuttal · Authors · 2024-08-16
>
> Thank you for your positive review recognizing the importance of the problem and our contribution to the community. We appreciate your concerns about making sure all the details of the experiment are provided in the text of the paper as well as in the data release itself.
>
> We believe all of this is very easily addressed during the revision process, so we are hopeful that you will consider raising your score to above the acceptance threshold after these additions.
>
> > [better] description of the data collection procedures and analysis of obtained data… as well as description of the experimental setup.
>
> Many experimental details were discussed on pages 8, 9, and 10 of the Supplement (see below) as part of the recommended Datasheets for Datasets section. This was a convenient place for us to include this information, as it is quite lengthy. In light of your comments, we will include the details you mention in the final version of the main text.
>
> * Re: ground truth data
>     * Supplemental Section 2.17, 2.22
>
> * Re: species typical vocalizations
>     * Supplementary Section 2.16 Edison Dataset
>     * Supplementary Section 2.16 Earbud Dataset
>
> * Re: dataset collection, analysis, and experimental setup
>     * Supplementary Sections 2.16. 2.17, 2.22
>
> > how long were the vocalizations
>
> We have attached a graph (Figure 1) in the rebuttal PDF detailing the vocalization lengths for each dataset.
>
> > properties of the random walks and real trajectories
>
> The most important detail here is the distribution of spatial position at which a sound event occurs. We agree it would be nice to visualize these positions as a 2D histogram / heatmap as a supplemental figure; we have included this in the rebuttal PDF (Figure 2).
>
> This is of course just a summary plot -- the exact position of each sound event is available to benchmark users. It is straightforward for users to run their own analyses, including generating the plot referenced above.
>
> > what were the species typical vocalizations
>
> We have shown example vocalizations in Fig 1C and Supplemental Fig 1A. Beyond that, we think the best we can do is refer the reader to established literature for characterization of mouse and gerbil vocal repertoires ([Heckman et al. 2016](https://doi.org/10.1016/j.neubiorev.2016.03.029); [Peterson et al. 2023](https://elifesciences.org/reviewed-preprints/89892)).
>
> Of course, the raw audio is available to all benchmark users, so they can generate whatever summary analysis plots they need.
>
> > how exactly the ground turn positions were obtained
>
> Details about ground truth position determination are detailed in Supplemental Section 2.17, 2.22. In brief, we used either 1.) a supervised keypoint detection algorithm called SLEAP to track the sound source (nose of the rodent, tip of the earbud) or 2.) using OpenCV, we convert the image to hsv space (`cv2.cvtColor`), threshold to isolate the speaker location, and compute its centroid (`cv2.moments`). We will put the exact code will be in our github repo during the revision so all of this is completely reproducible.
>
> > Some analysis of the accuracy of ground truth locations would also have been valuable.
>
> This is a great suggestion. Thus far, we have manually verified that ground truth locations were reasonable estimates of the true sound source position (see Supplementary Section 2.22). Formal quantification of the expected error using SLEAP are detailed in their [paper](https://www.nature.com/articles/s41592-022-01426-1), which describe <1mm error in keypoint. Finally, we note that the raw images are included in the benchmark so that users can always spot check any given data instance.
>
> We think it is a great idea to better formalize our manual validation of the ground truth locations. We will have three separate authors spot check 100 random frames from each dataset and provide a statistical quantification of the resolution (as well as human-to-human variability). We note that our best models achieving <1cm resolution on the SSL task suggests that the label noise is (at the very least) below this threshold.
>
> > Also the description of the baseline experiments is a bit thin, especially the description of Task 2 (vocalization attribution) would have deserved a bit better motivation and analysis.
>
> We will include an updated description/motivation in the revision. The basic premise is to see whether sound source localization (Task 1) is at a fine enough spatial resolution for us to be able to confidently attribute vocal calls to pairs of interacting animals. When we record the behavior from two pairs of animals, there is no ground truth measurement of the sound source. However, *we know that the vocal call must have come from one of the two animals*. If the 95% confidence interval produced by the model contains either zero animals or both animals, the model is unable to attribute the call. The frequency of these "missed attributions" is a concrete way to benchmark models against each other even thought we don't have a ground truth measurement of which animal produced the call. Let us know if more explanation would be helpful.

---

> > ### Comment · Reviewer_XE72 · 2024-08-28
> >
> > Thank you, these are useful additions. I will update my score.

---

### Official Review · Reviewer_8UsB · 2024-07-18
**Dataset and benchmark for sound source localization in rodents**

**Rating:** 7
**Confidence:** 4
**Correctness:** The authors correctly claims the data…

**Review:**

See the below sections.

**Strengths:**

The dataset and benchmark fill the gap between the acoustic machine learning and neuroscience communities.
The rodent dataset of video and multichannel audio is original, and the speaker, robot, earbud, and solo/dyad animal datasets are well designed for SSL evaluation in rodents.

**Additional Feedback:**

Here are several possible references.
* "Sound Imaging of Nocturnal Animal Calls in Their Natural Habitat," Mizumoto+, 2011. (A frog call paper)
* "Deep Learning Based Audio-Visual Multi-Speaker DOA Estimation Using Permutation-Free Loss Function," Wang+, 2022. (Multi-modal DOA (MDOA) for speaker diarization may be related to the third paragraph of the discussion section.)
* "Mind the Domain Gap: a Systematic Analysis on Bioacoustic Sound Event Detection," Liang+, 2024. (Not localization but another bioacoustic paper.)

**Clarity:**

This paper is easy to understand.
Even if we don't have any specific knowledge about rodents, the paper explains the necessary information.

**Documentation:**

The authors provides sufficient detail regarding the data collection process and availability.

**Ethics:**

There are no ethical concerns associated with this submission.

**Limitations:**

See the Opportunities For Improvement.

**Opportunities For Improvement:**

While the image source method (ISM) and 1D CNN are reasonable choice for simulation and network architecture, the authors could use another simulation and network architecture in future.

The reference could be for another simulation and network architecture.
* "RIR-in-a-Box: Estimating Room Acoustics from 3D Mesh Data through Shoebox Approximation," Kelley+, 2024.
* "A Four-Stage Data Augmentation Approach to ResNet-Conformer Based Acoustic Modeling for Sound Event Localization and Detection," Wang+, 2023.

**Relation To Prior Work:**

It is well discussed how this work is different from previous contributions.

**Summary And Contributions:**

The authors present a new dataset and benchmark called Vocal Call Locator Benchmark (VCL) for sound source localization (SSL) in rodents.
To understand how animals process acoustic information, SSL help us to determine the senders and receivers of acoustic information in social interactions.
The VCL serves datasets and benchmarks to evaluate SSL algorithms in the domain of bioacoustics.
The dataset consists of synchronized video and multi-channel audio recordings of 767,295 sounds with annotated ground truth sources across 9 conditions.
The SSL evaluation is conducted on simulated data and real data.

---

> ### Author Rebuttal · Authors · 2024-08-16
>
> Thank you for the positive review and suggested citations / future work. As you noted, we wanted to start with the simplest baseline model (1D CNN with temperature scaling for uncertainty calibration) and simulation method (image source method).
>
> We comment on each of the references you mentioned below.
>
> > "RIR-in-a-Box: Estimating Room Acoustics from 3D Mesh Data through Shoebox Approximation," Kelley+, 2024.
>
> This paper aims to estimate the room impulse response (RIR) from its geometry. This is potentially very relevant as a way to help DNNs trained on the SLL task in one environment generalize to a different environment with known geometry. We will certainly consider this approach and try to work this into our discussion of future work. For now we think the main focus should be to perform SSL well in an environment with a known RIR.
>
> > "A Four-Stage Data Augmentation Approach to ResNet-Conformer Based Acoustic Modeling for Sound Event Localization and Detection," Wang+, 2023.
>
> This paper introduces two novel data augmentation methods, audio channel swapping (ACS) and multi channel simulation (MCS), as part of a four-step augmentation pipeline for training deep neural networks for sound source localization and detection (SELD). Additionally, it appends Conformer modules to a ResNet to create an architecture well-suited to the SELD task. While ACS, as described in this paper, cannot be used in any of our environments, as our microphone arrays are not spherical, the concept of leveraging symmetries in the arrays to create augmented data/label pairs is applicable. Additionally, the MCS approach of factorizing recordings into spatial and spectral components which can be freely recombined is a promising augmentation for smaller datasets which we can explore further for our use case. Finally, we can include the ResNet-Conformer architecture as another benchmark for comparison.
>
> > "Sound Imaging of Nocturnal Animal Calls in Their Natural Habitat," Mizumoto+, 2011. (A frog call paper)
>
> This paper introduces a creative solution to SSL, whereby microphones convert audio signal into light when it detects a nearby sound. Using a video camera, users can then determine the location and timing of vocalizations occurring within a 30cm radius from each sensor. It’s possible that a related system could be adapted for use in rodents, however the main limitation is that rodents frequently vocalize during close-proximity interactions (~1 cm). To localize at such a small spatial scale, we would need many densely packed ultrasonic microphones, which in theory is possible, but in practice is costly and computationally intensive. Nonetheless, this is an interesting approach and something to consider seriously moving forward. Thank you for the suggestion!
>
>
> > "Deep Learning Based Audio-Visual Multi-Speaker DOA Estimation Using Permutation-Free Loss Function," Wang+, 2022.
>
> As suggested, we will cite this in the third paragraph of the discussion on multi-modal approaches. Thanks for the pointer.
>
> > "Mind the Domain Gap: a Systematic Analysis on Bioacoustic Sound Event Detection," Liang+, 2024.
>
> This is an interesting paper which experiments with using few-shot learning on a bioacoustic sound event detection task. Although this paper doesn’t focus on sound source localization, the finding that few-shot learning can improve task performance is something that we can experiment with in our own tasks. Moreover, the sensitivity of the task performance to which specific acoustic feature set they used is compelling. We could experiment with alternate featurization of our input data for example, which currently only operates on raw audio.

---

> > ### Comment · Reviewer_8UsB · 2024-08-30
> > **Response to rebuttal**
> >
> > Thank you for your detailed comments on my suggested references. I hope these references and comments will support the part of discussion. My review remains positive.

---

### Official Review · Reviewer_bexs · 2024-07-23
**How to Not Lose Your Earbuds**

**Rating:** 7
**Confidence:** 4
**Clarity:** The paper is clearly written.

**Review:**

As much as it pains me to say it, this study seems a bit too specific to act as a general benchmark. It seems like a useful set of data for evaluating localization of rodent vocalizations within the author's test apparatus, but it's unclear whether I should expect good results on this benchmark to carry over to new environments, or across taxa.

The ultimate goal is of more general interest, however: Can we diarize animals, even in a fixed environment? But since this is too hard to groundtruth, we have a number of very precisely groundtruthed SSL tasks, which only get at the behavioral question elliptically. It would be interesting to look at the SSL evaluation specifically through the lens of the ultimate goal: Diarization and classification of different rodent vocalizations. How does vocalization type and frequency impact localization quality? That would make a great study!

Likewise, it would be helpful to breakdown or better understand the factors in localization performance, for example looking at the covariance of error with vocalization frequency, centrality in the test chamber, etc.

With so much synthetic data, it would seem trivial to test sounds from a much wider variety of taxa. It would also be interesting to train on synthetic data only and evaluate the trained model on real-world data.

I am also kinda surprised that surgically attaching an earbud speaker to a rodent to get groundtruth locations is easy enough, but that social vocalizations can't be localized? I would expect that attaching a recording earbud/tag to each rodent and allowing them to interact would allow easier identification of which animal is vocalizing.

Overall, this work feels like groundwork for an interesting study, rather than a general-purpose benchmark.

**Strengths:**

The paper has a variety of experiments which address different weakneses in methods of generating groundtruth data for the study. The application of localization in bioacoustics in a lab setting is to my knowledge pretty novel.

**Additional Feedback:**

None

**Correctness:**

The paper seems correct. The evaluation is good within its very specific domain.

**Documentation:**

The data is readily available from the linked website, which exists.

**Ethics:**

Somehow the NeurIPS ethics guidelines completely misses animal study ethics! Please include a statement on how questions around animal study ethics were addressed. (eg, surgical implants for playback experiments.)

**Limitations:**

The authors have discussed the limitations of the work.

**Opportunities For Improvement:**

It would be great to see:
* Analysis of the relationship between vocalization type/qualities and localization error.
* Synthetic-to-real transfer quality.
* Use of the speaker-based systems to study the localization of a wider variety of taxa.
* More direct/ambitious attack on the diarization problem.

**Relation To Prior Work:**

Note that there are some existing attempts at localization in the wild using microphone arrays:
https://onlinelibrary.wiley.com/doi/full/10.1002/ece3.6216

https://www.frontierlabs.com.au/post/acoustic-localisation

And this crazpants dataset of thirty bats tracked with IR cameras in situ:
https://thejasvibr.github.io/ushichka/

**Summary And Contributions:**

This paper encapsulates a number of sub-datasets for localizing rodent vocalizations. Groundtruthing actual vocalizations, especially for diarization studies, is difficult. Groundtruth is generated through a number of playback experiments, including fixed speaker, speakers mounted on robots, earbud speakers surgically mounted on rodents(!), and actual rodent vocalizations.

---

> ### Author Rebuttal · Authors · 2024-08-16
>
> Although we agree with many of your points, we hope to convince you that our current manuscript is enough to advance the field and is in line with prior neuro benchmarks published in the D&B track.
>
> > It's unclear whether I should expect good results … to carry over to new environments, or across taxa
>
> The benchmark was collected across two labs (one at Princeton and one at NYU), spanning 3 test environments and 2 species with distinct vocal repertoires. In the revision, we will emphasize that gerbil vocalizations cover a particularly broad range of the sonic and ultrasonic spectrum (1-80 kHz) and we have every reason to believe that the benchmarks will generalize to many other vertebrate taxa whose calls contain very similar spectrotemporal and harmonic features (Sainburg et al., 2020; Peterson et al., 2023)
>
> The core criticism here is fair, but could be leveled at a majority of benchmarks. Animal pose tracking is notoriously sensitive to subtle changes in lighting (for example). The NeurIPS D&B track nonetheless publishes papers in this domain with data from single species and single labs (Sun et al. 2021; Marshall et al. 2021) and these lead to important scientific advances.
>
> This criticism could also be applied to many existing SSL benchmarks in the acoustic ML literature, which are mostly targeted at a handful of human-specific environments.
>
> Finally, as we explain below, our data could be used in conjunction with non-neuro benchmarks.
>
> > we have a number of very precisely groundtruthed SSL tasks, which only get at the [diarization problem] elliptically
>
> SSL and diarization really seem to go hand-in-hand in our setting. Diarization of human speech leverages acoustic feature differences between speakers, but many mammals (particularly mice) have variable vocalizations (Sainburg et al. 2020, Fig 8O), complicating this approach.
>
> Moreover, SSL is still a foundational task in acoustic ML. Thus, a plausible use case is for ML researchers to use our datasets in conjunction with human-specific benchmarks (e.g. L3DAS, LOCATA, STARSS23). Testing across these domains would be a very strong test of generalization!
>
> > a recording earbud/tag on each rodent … would allow easier identification of which animal is vocalizing
>
> We are continually investigating new experimental solutions. The upshot thus far is that speaker identity is ambiguous when attaching microphones; e.g. when rodents are face-to-face there is complex cross-talk across microphones.
>
> Importantly, the field wants to diarize animals *without implanting anything on them*. So while this could be helpful as part of a benchmark, we still need to develop SSL techniques that don’t rely on this.
>
> >  it would be helpful to breakdown or better understand the factors in localization performance
>
> Great suggestion. We have done various analyses along these lines, but they didn’t make it into the submission. As one might expect, localization is more challenging for low sound level and high frequency calls. Performance tends also to be better in areas with more training data. We are happy to answer additional questions. It is very easy for us to include these analyses in our revision.
>
> > With so much synthetic data, it would seem trivial to test sounds from a much wider variety of taxa.
>
> While it is easy to investigate more taxa purely in simulation, this does not transfer to the real world (see below).
>
> > train on synthetic data only and evaluate the trained model on real-world data.
>
> We have done this and found that models perform poorly. This can be inferred from Fig 2D. In the revision, we can extend the x-axis here to include models trained on 100% simulated data. Right now it cuts off at 70% because models start performing very poorly beyond this. Our efforts are a starting point to improve synthetic-to-real transfer.
>
> > groundwork for an interesting study, rather than a general-purpose benchmark
>
> We appreciate this perspective, but it is rare for neuro papers to contain benchmark data that does not test a biological hypothesis.
>
> We could potentially publish a specialized “methods paper” that focuses only on the deep network model rather than the datasets. This is not ideal because the data is highly valuable, being both labor-intensive to collect and requiring expensive equipment. Prior works describe valuable SSL technology but have not provided the field with large-scale benchmark data (Warren et al., 2018; Sterling et al., 2023).
>
> We would like to find a venue (such as NeurIPS D&B) which will advertise these datasets as a resource to spark more research. As mentioned above, SSL is a very actively studied topic in acoustic ML and we hope to attract interest from at least some portion of that field. This would be unlikely in a neuro journal.
>
> > Opportunities for improvement:
> > * Analysis of the relationship between vocalization type/qualities and localization error.
>
> Our revision will address this as explained above. Let us know if you’d like more detail.
>
> > * Synthetic-to-real transfer quality.
>
> Synthetic-to-real transfer quality is somewhat poor, though we see slight performance gains with 30:70 simulated-to-real mixtures of training data. We include RIRs and environment specs to encourage improvements from users. Developing the optimal solution is not the goal for this initial benchmark release, so we don’t see this as an opportunity for improvement.
>
> > * Use of the speaker-based systems to study the localization of a wider variety of taxa.
>
> We deliberated at length whether to promise this (e.g. with bird calls), but this is very labor intensive and we are uncertain this would improve the paper. If you feel very strongly that this would make the difference in your decision, please let us know.
>
> > * More direct/ambitious attack on the diarization problem.
>
> We plan to investigate this in future work. We believe SSL is closely linked to speaker identification in this application setting, so our benchmark is a valuable first step.

---

> > ### Author Response · Authors · 2024-08-16
> > **References in Rebuttal**
> >
> > * Marshall, J. D., Klibaite, U., Aldarondo, D. E., Olveczky, B., & Timothy, W. D. The PAIR-R24M Dataset for Multi-animal 3D Pose Estimation. *Advances in neural information processing systems &mdash; Datasets and Benchmarks Track*.
> >
> > * Peterson R, Choudhri A, Mitelut C, Tanelus A, Capo-Battaglia A, Williams AH, Schneider DM, Sanes DH (2023). Unsupervised discovery of family specific vocal usage in the Mongolian gerbil. *eLife12:RP89892*.
> >
> > * Sainburg, T., Thielk, M., & Gentner, T. Q. (2020). Finding, visualizing, and quantifying latent structure across diverse animal vocal repertoires. *PLoS computational biology*, 16(10), e1008228.
> >
> > * Sterling, M. L., Teunisse, R., & Englitz, B. (2023). Rodent ultrasonic vocal interaction resolved with millimeter precision using hybrid beamforming. *eLife*, 12, e86126.
> >
> > * Sun, J. J., Karigo, T., Chakraborty, D., Mohanty, S. P., Wild, B., Sun, Q., ... & Kennedy, A. (2021). The multi-agent behavior dataset: Mouse dyadic social interactions.*Advances in neural information processing systems &mdash; Datasets and Benchmarks Track*.
> >
> > * Warren, M. R., Sangiamo, D. T., & Neunuebel, J. P. (2018). High channel count microphone array accurately and precisely localizes ultrasonic signals from freely-moving mice. *Journal of neuroscience methods*, 297, 44-60.

---

> > ### Comment · Reviewer_bexs · 2024-08-21
> >
> > Thank you for the thoughtful response. This helps clear up a number of questions, and I'll be raising my score for the paper accordingly.
> >
> > Adding a note on the synthetic data model performance would be helpful for context. I believe there is an opportunity to attach an appendix or supplemental information, which could be a good way to share the analysis of performance factors.
> >
> > I'm largely sympathetic to the difficulties of publishing ML-oriented datasets in non-ML fields, so this seems like a good place to provide a home.

---

> > > ### Author Response · Authors · 2024-08-21
> > > **Official Comment by Authors**
> > >
> > > Thank you for your encouragement. We will be sure to expand our discussion of the synthetic data simulation + modeling in the final appendix.

---

### Official Review · Reviewer_ioCn · 2024-07-24
**Proposed dataset and benchmark results for localizing rodent vocalizations from audio is well put together, while the paper's writing could be improved.**

**Rating:** 7
**Confidence:** 4

**Review:**

This is a relevant paper for the NeurIPS Datasets & Benchmarks track, with an application domain in the intersection of bioacoustics, animal cognition, and neuroscience. The authors argue for a lack of large-scale datasets of rodent vocalizations, especially for training methods that rely on SSL. The dataset itself is carefully constructed and is also complemented by synthetic data. Two tasks are benchmarked against the dataset, one on sound source localization and the other on vocalization attribution - in both cases relevant baseline models are evaluated.

Overall this is a good paper which does address an existing gap in the aforementioned communities. In terms of the paper reaching its audience, I have some concerns since the writing and prose of the paper is discipline specific and might be deemed inaccessible by a generalist NeurIPS reader. As a reviewer whose work has covered bioacoustics in the past I was able to follow the paper but I can see how the paper's impact can be diminished by the way the dataset and methods are presented.

As a final note, since this is a dataset involving data collection from real rodents, I would have expected to see a clear ethics statement on data collection from animals at the end of the paper. I was surprised that this statement was absent, although was happy to see that this is included in the supplementary PDF. Given the importance of ethics approvals when dealing with data collection from animals, I would expect to see an abbreviated statement or at least a reference on that statement from the main paper.

**Strengths:**

* Well constructed and new large-scale dataset for benchmarking self-supervised learning algorithms in rodent vocalizations.
* Appropriate baseline methods chosen, and appropriate experimental setup.

**Additional Feedback:**

No further feedback to add.

**Clarity:**

As mentioned above, the paper is written in a discipline specific way (geared towards a neuroscience / animal cognition / animal behavior / bioacoustics audience) and can be significantly rewritten as to cater for a generalist audience at NeurIPS that would be interested in working with new datasets and benchmarks.

**Correctness:**

To my understanding, all claims made in the submission are justified. The dataset is constructed in a sound way, and similarly the methods used and experimental setup are technically correct.

**Documentation:**

There is a rich supplementary document that further details data collection and availability. Ethical and responsible use are also covered.

**Ethics:**

As mentioned above, the main paper could include a clear mention on ethics approvals related to data collection from animals. This is included in the supplementary PDF.

**Limitations:**

The authors do carefully consider limitations of this work, and attempt to consider potential negative social impact (for which they do not identify any issues).

**Opportunities For Improvement:**

* Writing can be simplified and made more accessible for a generalist NeurIPS audience.
* Include a mention on ethics approvals in the main paper.

**Relation To Prior Work:**

Related work has been cited and discussed - no additions to make.

**Summary And Contributions:**

* New large-scale dataset for benchmarking self-supervised learning algorithms in rodent vocalizations.
* Benchmark results on both real and simulated data.

---

> ### Author Rebuttal · Authors · 2024-08-16
>
> Thank you for the positive review and constructive feedback. We believe we can easily incorporate your two suggestions. Please let us know if our revision plan below meets your approval.
>
> > Include a mention on ethics approvals in the main paper.
>
> Yes, we should have done this in the first place rather than in the supplement. We will include the following statement in the main text of our revision: “All procedures related to the maintenance and use of animals were approved by the University Animal Welfare Committee at New York University and Princeton University. All experiments were performed in accordance with the relevant guidelines and regulations.”
>
> > Writing can be simplified and made more accessible for a generalist NeurIPS audience.
>
> We agree and will improve the writing. To do this effectively we will distribute our paper to several colleagues at NYU in computer science and machine learning who specialize in acoustics (but not in biological applications), asking them to identify specific paragraphs and sentences they find unclear. We will also distribute this to several graduate students who work on general machine learning applications (not acoustics) to get similar feedback. We want our paper and benchmark to be as broadly accessible by experts and young investigators.

---

> > ### Comment · Reviewer_ioCn · 2024-08-28
> >
> > Thank you for the clear response, much appreciated especially that the ethics mention will be included in the main paper.

---

### Author Rebuttal · Authors · 2024-08-16

We are grateful for the feedback from all reviewers. We summarize highlights below and provide detailed responses to each reviewer individually.

---

Two positive reviews (`ioCn = 7`, `8UsB = 7`) left constructive suggestions which we will incorporate. Both provided high confidence scores and accurately summarized the motivation and technical content of our paper.

---

One borderline negative review (`XE72 = 5`) identified some methodological information that is missing from the paper. We provide these details in our individual response and will incorporate them in our revision. Our **rebuttal pdf (see attached)** contains two example figures that we will include in our revision. Additionally, our original submission included three pages of experimental methods in the supplement; parts of this may have been inadvertently overlooked, so please let us know if details are missing from that section. Importantly, `XE72`’s comments were generally very positive about our study and its relevance to NeurIPS.

---

We received one negative review (`bexs = 4`), which was constructive and written in good faith. The review recognizes the potential of our study, stating that our goals are of “general interest.” As highlighted in our individual response to `bexs`, we can incorporate at least one of their helpful suggestions for improvement.

Here we want to highlight what we believe is a central misunderstanding—`bexs` asks that our results generalize beyond “the author's test apparatus” and “across taxa.” We emphasize that the **data was actually collected across two labs (one at NYU and one at Princeton)** and across two taxa (mice and gerbils). Our results also encompass three different physical environments with unique microphone configurations as well as data from speakers (vocal calls + synthetic sounds) and animal-generated vocalizations. Taken together, we believe this provides a much stronger test for generalization than similar work accepted to the D&B track. For example, impactful work on pose estimation by [Sun et al. (2021)](https://openreview.net/forum?id=NevK78-K4bZ) and [Marshall et al. (2021)](https://datasets-benchmarks-proceedings.neurips.cc/paper_files/paper/2021/hash/1ff8a7b5dc7a7d1f0ed65aaa29c04b1e-Abstract-round1.html) used one apparatus from a single lab and in a single species.

We appreciate `bexs`’s suggestion that our work could be a “scientific study” instead of a benchmark, but it is very rare for neuroscience journals to publish data from multiple species in a single paper or data that does not address a biological hypothesis. At the same time, we do not believe that further broadening the scope of the benchmark at this stage (e.g. by including new recordings from non-rodent species or recordings from the wild) would significantly improve the impact of the paper.

Ultimately, since this is the first major public dataset released in this application area, we aimed to strike a balance between a study that is over-optimized to a single lab and an extremely broad benchmark with uncontrolled parameters. We settled on a benchmark that includes multiple rodent species recorded across multiple labs and on different experimental setup. We hope that the reviewers agree this is a good first step.

---

### Author Response · Authors · 2024-08-26
**Summary of discussion thus far**

Leading into the last week of the rebuttal period (ending on Aug 31), we want to take the opportunity to thank the reviewers once again for their feedback on our manuscript. As of Aug 26, the reviewer scores are `7 - 7 - 7 - 5` &mdash; i.e. three "Good paper, accept" votes and one "Marginally below acceptance threshold" vote.

We particularly want to thank reviewer `bexs` for engaging with our rebuttal and raising their score to a seven. We have not yet heard from the other three reviewers but are ready to respond to any last minute additional questions or comments throughout the week.

We hope to hear that our response to reviewer `XE72` (the remaining "below threshold" score) is satisfactory. Our reading of this review is that they liked the motivation and execution of our study, but requested additional methodological details about the experimental setup. In particular, we liked their suggestion to include a formal analysis of the label noise in our benchmark (i.e. how accurate is the spatial location of the "ground truth" sound source). We are currently working on this as described in our response.

Overall, we believe that the comments by `XE72` can be addressed in our revision. Indeed, a couple of the requested items were already included in the supplemental appendix of our original submission. None of the other reviews mentioned serious concerns about missing methodological details that rise to the level of rejecting the manuscript. For these reasons we are hopeful for a positive outcome. We politely ask that the reviewers let us know of any outstanding major concerns within the next few days so that we may have a chance to respond and rectify them.

---

### Decision · Program_Chairs · 2024-09-26

**Decision:**

Accept (Poster)

**Comment:**

I would like to commend the authors for their diligent work and the constructive engagement they demonstrated both before and after submission. The paper has garnered strong support from the reviewers, with final scores of 7-7-7-6, all recommending acceptance. The authors effectively addressed the key concerns raised, and their thorough rebuttal clarified and strengthened the manuscript further.
This meta-review will summarize the paper's strengths in terms of quality, originality, and significance, as well as outline the main pros and cons highlighted in the reviews, rebuttal, and discussion. And it is assumed that all commitments made by the authors will be fulfilled in the camera-ready version.

## Quality
The main contribution of this work is a large-scale dataset for benchmarking both sound source localization (SSL) and diarization methods applied to rodent vocalizations in lab environments by using multi-channel audio. This work has been recognized for filling a gap between acoustic machine learning and neuroscience according to reviewer [8UsB]. The dataset is large and well-designed/well-constructed [ioCn, 8UsB], featuring vocalizations from two species of rodents (mice and gerbil) and non-animal sounds produced by other agents (robot, speaker, and earbud) simulating animal-like vocalizations or synthetic sounds. The dataset is also cross-lab, as it was recorded in 9 environments across 2 different labs. The authors clarified in their rebuttal that the data collection across multiple labs and taxa (mice and gerbils) offers a stronger test for generalization than similar work accepted to the D&B track. The work places significant emphasis on ground truth generation [bexs], a challenging task accomplished using video signals, a keypoint detection algorithm, and manual curation. The benchmarking methods were appropriately chosen [ioCn], and the evaluation procedure was sound [bexs].


## Clarity
According to most reviewers (3 out of 4), the paper was clearly written and self-contained. However, one reviewer suggested that the prose might pose a barrier to the broader NeurIPS audience [ioCn]. The authors promised to revise the manuscript with feedback from peers (specialized in acoustics outside biological applications) to make the text more accessible. Also, [XE72] noted that the description of the diarization task (Task #2) was somewhat ‘thin’ before the revision. The authors provided the necessary details in their rebuttal and will include it in the manuscript (if not already present, as some details were in the supplementary material and overlooked).


## Originality

The application of SSL in bioacoustics is novel [bexs], and the dataset was constructed from scratch. The method of establishing ground truth for multi-channel audio using video for this type of data is also original, as is the inclusion of synthetic sounds (speaker, robot, and earbud) alongside gerbil/mice vocalizations in solo/dyadic settings [8UsB]. The authors responded to reviewer concerns by emphasizing that their dataset was designed to balance specificity and generalization, addressing both the need for controlled experimental setups and broader applicability across multiple species and labs.


## Significance

Benchmarking SSL and diarization of rodent vocalizations might complement human-based benchmarks (e.g., LOCATA or L3DAS). This dataset provides an alternative, more challenging benchmark due to the broader spectrum of frequencies compared to human speech. Additionally, as the authors noted, it enables cross-domain evaluation to assess the generalization of SSL/diarization methods.


## Pros and Cons
**Pros:**
-	Novel dataset for benchmarking SSL and diarization methods. The rodent vocalizations cover a broader spectrum of frequencies than human speech from existing human-centered benchmarks, making this a challenging and complementary benchmark.
-	The dataset was collected across multiple labs (cross-lab) and includes multiple rodent species (cross-taxa), enhancing generalizability.
-	Well-designed experiments and thorough evaluation.
-	No major ethical concerns; ethics approvals were granted by the authors' institutions (now mentioned in the main paper).
**Cons:**
-	The evaluation of Task 2 lacks associated ground truth, making it more cumbersome to assess (see the authors' last response to [XE72] for details).